# Exports, Misallocation, and Total Factor Productivity of Furniture Enterprises

**Chang Xu [1], Jianbing Guo [2], Baodong Cheng [1,\*] and Yu Liu [3,4,\*]** 

1    School of Economics and Management, Beijing Forestry University, Beijing 100083, China
2    School of Agricultural Economics and Rural Development, Renmin University of China,
      Beijing 100872, China
3    Institutes of Science and Development, Chinese Academy of Sciences, Beijing 100190, China
4    School of Public Policy and Management, University of Chinese Academy of Sciences, Beijing 100049, China
\*    Correspondence: baodong@bjfu.edu.cn (B.C.); liuyu@casipm.ac.cn (L.Y.)

**Abstract:** With the increase in labor costs in China and the tremendous changes in the international trade environment, upgrading the total factor productivity of Chinese furniture export enterprises faces a great challenge. Lots of studies have explored the interaction of exports or misallocation on the total factor productivity (TFP) of furniture enterprises, however, there is little knowledge on the impact and interaction of both exports and misallocation on the TFP. Based on panel data of Chinese furniture enterprises, this paper measures the TFP and the distortion of labor and capital resources in Chinese furniture enterprises. A two-way fixed-effects model is used to analyze the impact of exports and misallocation on the TFP of Chinese furniture enterprises. The paper reveals several important findings. First, the TFP of Chinese furniture export enterprises is lower than that of non-export enterprises, this phenomenon is called the "export–productivity paradox". Chinese furniture export enterprises are processing trade-oriented and labor-intensive enterprises at the low end of the value chain, exports have a negative effect on improving the TFP of furniture enterprises in the short term. Second, the distortion of labor and capital resources in Chinese furniture enterprises promotes improvements to the TFP of furniture enterprises rather than reducing the TFP of furniture enterprises. Last but not the least, we find that misallocation has a positive moderating effect on exports and can weaken the negative impact of exports on TFP by the "forced mechanism", which is that the higher the distortion of the misallocation, the higher the cost of acquiring capital and labor, and enterprises are forced to enhance their productivity when facing market competition, thus promoting improvements to the TFP of furniture enterprises.

**Keywords:** exports; misallocation; total factor productivity; furniture enterprise

---

## 1. Introduction

China is an important furniture producer and exporter. As China's furniture exports exceeded those of Italy in 2004, it is the world's largest furniture exporter. In 2018, the total exports of the Chinese furniture industry totaled \$53.685 billion, a year-on-year increase of 5.7%, with a high market share. Although the Chinese furniture industry has developed rapidly over the past 20 years, it has evolved from a traditional handicraft industry to an industrial automation production with advanced technology and equipment. Chinese furniture enterprises are still labor-intensive enterprises, and cheap labor is still the main source of their competitive advantage. As the laboring-age population turned to negative growth after 2010, and the labor prices continue to rise [1], the per capita wage of China's manufacturing industry continues to grow at a high rate, and the growth rate has been maintained at more than 10%. In 2013, the per capita wage of China's manufacturing industry was

46,431 yuan, more than three times that of 2003. In 2005, US wages were 20 times that of China, and in just five years, it was reduced to 10 times. As southeast Asian countries, such as Vietnam and Malaysia, have lower wages than China, some Chinese furniture companies have begun to move to Southeast Asia for lower costs. In 2015, the labor productivity of China's labor was only 40% of the world average. The advantage resource of cheap labor is quietly dissipating, and furniture enterprises must enhance the TFP to get new competitive advantages. The TFP refers to the factors that promote economic growth in addition to capital and labor factors. The development of the furniture industry is uncertain due to the current complex international situation. In July 2018, the Office of the US Trade Representative announced a list of 10% tariffs on the imports of 200 billion US dollars from China. In this list, the names of furniture goods accounted for two pages. The two pages included almost all furniture products exported by China to the United States, such as wooden furniture and wooden toys. In September 2018, China levied 5–25% on retaliatory taxes on timber imported from the United States. On 5 May 2019, Trump, the US President, issued a tweet that would increase the tariff on China's $200 billion to US products from 10% to 25% on 10 May. On 5 August 2019, Trump, the US President, issued a tweet that would increase the tariff of 10% on China's $300 billion to US products. However, Chinese furniture export enterprises are more engaged in international OEMs (original equipment manufacturer), relying on foreign companies' outsourcing orders and raw materials, and are forced to attach to multinational companies. Chinese furniture export enterprises are located at the low end of the value chain, with less profit, mainly relying on the investment of China's labor resources and the development of processing trade, and the international trade environment has had some changes which will have a major impact on the operation of Chinese export furniture enterprises. In this context, studying the interaction between furniture exports, the distortion of labor resources, and TFP is of important practical significance to the development of furniture enterprises.

According to new–new trade theory, the heterogeneity of productivity exists in enterprises, and entering the export market requires entry costs, so that only high-productivity enterprises can enter the international market; the lower-productivity enterprises can only enter the domestic market, and the enterprises with high TFP will choose to export [2]. Romer found that exports can also increase enterprises' TFP [3]. However, the situation in China is not as predicted by the new trade theory. Some scholars have found that there are "self-selection effects" and "export learning effects" in corporate exports, and their research confirms the new trade theory [4–6]. However, some scholars have reached the opposite conclusion. They found that in China, the TFP of export enterprises is lower than that of non-export enterprises, and that one of the main reasons for this is the large number of processing trade enterprises. Processing trade is characterized by "two heads outside", i.e., the products produced are mainly used for export, and the cost of entering the international market is low [7–10]. According to the economic growth accounting framework proposed by Solow [11], the driving force for the improvement of TFP comes mainly from two aspects; one is the input of labor, land, capital, and other factors, and the other is from the allocation of elements. In recent years, scholars have also paid attention to the impact of resource mismatching on TFP efficiency. Researchers generally follow the theory proposed by Hsieh and Klenow [12], and measure the impact of resource mismatches on TFP using the production function regression method; however, they have arrived at different results. This may be due to the different types of enterprises examined. Based on the theory of enterprise heterogeneity, some scholars have reported that the impact of resource mismatching on TFP is negative [13–15], and some have discovered that it is positive [16]. China's furniture enterprises are typical in terms of the distortion of labor resources, and their exports are primarily composed of processing trade. Therefore, from the perspective of enterprise heterogeneity, this paper will explore whether exports from furniture enterprises support the new trade theory. From the perspective of economic growth and the development of furniture enterprises, TFP is an important driving force for economic growth, in addition to factor input. To promote China's economic development and the transformation and upgrade of furniture enterprises, we must improve the TFP; thus, it would be useful to measure it. Analysis and measurement of the changes in TFP are crucial to elucidating

the factors affecting it. Based on the current theoretical background, we aim to analyze the impact of exports and misallocation on the TFP of Chinese furniture enterprises.

The innovation and contribution of this paper are reflected in the following aspects First, we studied the influencing factors of the TFP of Chinese furniture enterprises. Most studies have focused on the TFP of all industrial enterprises in the whole industry, but research on a certain industry, especially for Chinese furniture enterprises, is rare. We found that the export behavior of furniture enterprises does not improve the TFP of furniture enterprises. On the contrary, the TFP of furniture export enterprises is lower than that of domestic enterprises. Most furniture enterprises are processing trade-oriented enterprises, and the large amount of processing trade reduces the entry cost into the international market. Most Chinese furniture enterprises are labor-intensive enterprises. When export production requires the use of rich resource belonging to the local the competition in the domestic market is more intense than in foreign markets and, the productivity of export enterprises is lower than that of non-export enterprises. We believe that the existence of a large number processing trade-oriented and labor-intensive furniture enterprises is the reason for this situation. The misallocation means that the factor market cannot allocate resources efficiently, resulting in distortion of factor prices in the factor market and deviation from actual prices [12]. The misallocation also promotes the TFP of furniture enterprises, which forces enterprises to increase the TFP to maintain competitive advantage. Therefore, this paper enriches the research on the factors affecting the TFP of furniture enterprises. Second, we explored the interactions between misallocation, exports, and TFP. Misallocation and exports are important factors affecting the development of micro-enterprises. Exploring the interaction between misallocation and TFP and analyzing the interaction between exports and the TFP of enterprises has been vast, but there are fewer studies which focus on the interactions between the three factors. When considering the effects of the distortions of labor resources and exports on TFP, the distortions of labor resources can cause negative effects from exports on the TFP of enterprises by the "forced mechanism", which is that the higher the distortion of the misallocation, the higher the cost of acquiring capital and labor, and enterprises are forced to enhance their productivity when facing market competition, and the distortion of capital resources can promote TFP to some extent. Our research can provide a new perspective on the impact of exports or misallocation on the TFP. Third, in terms of research methods, we have chosen a variety of methods to measure the TFP of enterprises, and compared the impact of various methods on the empirical results. We found that there are differences in TFP measured by different measurement methods, the TFP measured by the OP method is the smallest, while the TFP measured by the ACF-LP method is the largest. However, the use of different measurement methods has little effect on the results of empirical analysis. Our research can also provide scholars with an empirical basis for the selection of the methods to measure the TFP.

This paper is organized as follows. Section 2 proposes the research hypotheses and model. Section 3 describes the variables and data sources. Section 4 presents our empirical results. Finally, Section 5 outlines our discussion and policy suggestions.

## 2. Research Hypothesis and Setting Model

### 2.1. Research Hypothesis

According to the reference and Solow growth model, the TFP refers to the factors that promote economic growth in addition to capital and labor factors, which is an important index to measure productivity [11]. It is crucial to improve the TFP, but Young found that growth in China and East Asia's economy is mainly based on factors such as investment and labor. The rate of productivity growth is very slow, with an average annual growth rate between 2% and 3% [17]. Brandt used the China Industrial Enterprise Database to find differences in China's TFP from 1997 to 2007, but the average growth rate was 7.96% [18]. Studies on exports and TFP mainly focus on the following questions: What is the difference between TFP and export and non-export enterprises? Is there an "export–production efficiency paradox"? Do export enterprises exhibit the "self-selection effect" and "export learning

effect"? Scholars have reported the phenomenon of the export–productivity paradox in China. It is believed that most enterprises are mainly processing trade-oriented enterprises working in the Chinese export trade. Processing trade is characterized by "two heads outside", and enterprises are usually involved in manufacturing. Researchers believe that, in China's export trade, most enterprises are mainly processing trade-oriented enterprises. The products are mainly exported, and the large amount of processing trade reduces the entry cost into the international market, so enterprises do not need higher productivity, thus lowering the productivity of the entire export enterprise [17]. At the same time, most of China's industries are labor-intensive enterprises. When export production requires the use of rich local elements, the competition in the domestic market is more intense than in foreign markets, and enterprises entering the domestic market need to be more productive. Therefore, in the case of Chinese capital-intensive industries, the productivity of export enterprises is higher than that of non-export enterprises. For Chinese labor-intensive industries, the productivity of export enterprises is lower than that of non-export enterprises and, hence, we have an export–productivity paradox.

**Hypothesis 1 (H1).** *Chinese furniture export enterprises are processing trade-oriented and labor-intensive enterprises, and exports have a negative effect on improving the TFP of furniture enterprises. The furniture industry exhibits the export–productivity paradox.*

Since reform and opening policies have been implemented, the market economy has developed rapidly. However, the degree of marketization in China's factor market lags behind that of product markets. Land, labor, and capital are particularly subject to government intervention. The market cannot reasonably configure its resources, and misallocation can have a strong impact on TFP. Hsieh and Klenow reported that the higher the distortion of the capital resource, the higher the interest required by the enterprise and the cost of acquiring capital. Therefore, enterprises are forced to enhance their productivity when facing market competition, and the distortion of capital resources can promote TFP to some extent. Wang Yaxing found that misallocation enhances the threshold of productivity required for enterprises to enter the market. As Chinese labor prices rise, the costs to enterprises will increase further. However, if enterprises want to survive, they must increase their productivity and undergo transformation and upgrades. On the other hand, misallocation promotes TFP by a forced mechanism.

**Hypothesis 2 (H2).** *Misallocation affects enterprise efficiency, and misallocation in the furniture industry can promote TFP through the "forced mechanism".*

Based on the data from the China Industrial Enterprise Database, there are many papers on the relationship between resource mismatch and enterprise TFP, and the relationship between exports and TFP. However, there are relatively few studies on the relationship between the three. We conducted an empirical analysis based on the relationship between the three. As furniture companies are mainly labor-intensive enterprises, when discussing the relationship between the three, we focus on the distortion of labor factors. If hypotheses H1 and H2 are confirmed for furniture companies, exports have a negative impact on TFP, and labor mismatches will force companies to increase TFP. How will increasing labor mismatches affect furniture enterprise exports? Today, when China's demographic dividend is gradually weakening, it is crucial to answer this question so that we can transform and upgrade labor-intensive furniture enterprises that focus on processing trade. In the face of more serious labor mismatches, export companies may be forced to increase TFP. In recent years, with increasing labor costs in China, some processing trade enterprises have begun to shift to Southeast Asian countries with lower labor costs, and some enterprises have realized enterprise transformation and upgraded through reform, innovation, and machine substitution.

**Hypothesis 3 (H3).** *The distortion of labor resources has a regulating effect on exports, and labor distortion can further improve the total factor productivity of furniture enterprises through a forcing mechanism.*

## 2.2. Model

### 2.2.1. Benchmark Model

After calculating the TFP of the enterprise, based on the analysis and research hypothesis of H1, H2 and H3 above, we established the following benchmark model:

$$TFP_{it} = \beta_0 + \beta_1\varphi_{lit} + \beta_2\varphi_{kit} + \beta_3EXP_{it} + \beta_kX_{it} + \gamma_{it} + \omega_{it} + \varepsilon_{it}, \tag{1}$$

where $TFP_{it}$ is the total factor productivity, $\varphi_{lit}$ is the distortion of the labor resource, $\varphi_{lit}$ is the distortion of the capital resource, and $EXP_{it}$ refers to exports. If $\beta_1$ is positive, then hypothesis H1 is supported. $X_{it}$ represents other control variables, $\gamma_{it}$ is the enterprise fixed effect for controlling individual invariant elements, and $\omega_{it}$ is the enterprise fixed for controlling time invariant elements. $\varepsilon_{it}$ is the standard error term used to control other specific heterogeneous effects. (1) is used to explain the impact of exports and the distortion of labor and capital resources on the TFP. The specific impact can be determined based on the negativity of the coefficient after regression analysis. If the estimation coefficient passes the significance test, the above three hypotheses have a significant impact on TFP.

### 2.2.2. Moderating Effect

On the basis of the previous analysis, to verify hypothesis H3, we introduced a cross-item to identify the moderating effect of misallocation and exports in the model. As furniture is labor-intensive, the degree of distortion of labor resources in the furniture industry is greater than the degree of capital distortion, so we focused on the distortion of labor resources and established a model of the moderating effect, as follows:

$$TFP_{it} = \beta_0 + \beta_1\varphi_{lit} + \beta_2\varphi_{kit} + \beta_3EXP_{it} + \beta_4\varphi_{lit} \times EXP_{it} + \beta_kX_{it} + \gamma_{it} + \omega_{it} + \varepsilon_{it}. \tag{2}$$

The value of each variable in (2) is the same as in (1), where $\varphi_{lit} \times EXP_{it}$ is the product term of the distortion of the labor resources and exportation. We need to know the values of $\beta_1$, $\beta_2$, $\beta_3$, and $\beta_4$ to determine whether the distortion of labor resources has a moderating effect on exports. The specific results will be described later.

## 3. Data and Descriptive Analysis

### 3.1. Data source

The data covered China's manufacturing industrial enterprise data from 1999 to 2007. The database of China's manufacturing industrial enterprise was established by the State Statistics Bureau of China. The database includes all state-owned and non-state-owned industrial enterprises, and the annual sales of non-state-owned enterprises are greater than RMB 5 million. The database of China's manufacturing industrial enterprise accounts for 95% of total industrial output and includes long-term statistics and many statistical indicators. Hence, it is an important dataset for researchers studying the relationship between misallocation, exports, and TFP at the enterprise level. The database reports the industry classification, geographical location, number of people employed, type of registration, business status, and other economic indicators. However, there are some problems with the database, such as mismatched sample matching, abnormal sizes of variables, and missing index values, all of which require proper processing and analysis.

In this paper data were processed according to the following criteria [18–21]: (1) Exclusion of samples with abnormal index values; (2) Exclusion of samples from enterprises with missing or negative values in any of the intermediate investors, the average number of employees, the annual average balance of fixed assets, and the export delivery value; (3) To ensure the validity of the establishment time of the enterprise, the samples with enterprises established before 1949 were deleted, as were samples from enterprises whose enterprise age was less than 0; (4) The observations from

China's manufacturing industrial enterprise with negative industrial value were excluded, the lack of industrial added value and exports in 2004 would affect the measurement analysis, so the data from 2004 were excluded. If the added value was missing, it was calculated using the following formula: industrial added value = industrial gross output value – industrial intermediate input + value added tax for the current year [5]; (5) Different scholars have different methods for matching samples. This paper was mainly based on the method of "legal code + enterprise name"; (6) To unify the industry classification to make the data comparable and continuous, this paper converted the industrial code of China's manufacturing industrial enterprise from 1999 to 2007 into the industry code of GB/T4754-1994, according to China's National Economic Industry Classification. The furniture industry includes: wooden furniture manufacturing (2110), bamboo and rattan furniture manufacturing (2120), metal furniture manufacturing (2130), plastic furniture manufacturing (2140), and other furniture manufacturing (2190).

*3.2. Variable selection*

### 3.2.1. Dependent Variable

The dependent variable was the TFP of Chinese furniture enterprises. This paper used logarithmic values to represent the TFP. This paper was based on the following three methods: OP, ACF-OP, and ACF-LP. The TFP was calculated as follows: TFP = *ln*(Total Factor Productivity).

The methods for measuring the TFP of micro-enterprises mainly included the OLS, based on the Cobb–Douglas (CD) function, OP, LP, and ACF methods, and the De Loecker model for single products in non-competitive markets and the Beveren model for multi-product exporters [22–27]. The ordinary least squares (OLS ) estimation method is to obtain the parameter estimator under the condition that the sum of the squares of the objective function residuals is the smallest.The OLS method used the CD function to measure the TFP of enterprises, but was subject to endogenous problems, simultaneity deviation, sample selection bias, and missing price bias, so it was not a good estimation method. The LP method is very similar to the OP method, aimed at solving the problem of unobservable productivity impact and input factors. The LP method (2003) [22] studied by Levinsohn and Petrin used the intermediate input as a proxy variable to solve the problem of simultaneous deviation and endogeneity. Olley and Pakes (1996) [23] first used the semi-parametric estimator to measure micro-enterprise productivity,which is called the OP method.The OP method (1996)[23] assumed that the enterprise made investment decisions based on its current state of productivity, and used the current investment of the enterprise as a proxy variable to solve the problem of simultaneous deviation. The ACF method (2006) [27] studied by the Ackerberg et al. considers the assumption that labor is a free variable to be too strict.The ACF model is an extension of the OP model, which assumes that all of the input factors are state variables, and that productivity and input factors affect a company's investment decisions. The OP and LP methods have multicollinearity problems,then Ackerberg et al. propose a correction to the OP and LP methodologies,they presents a new approach, an application to real data and estimates on simulated data,which is called the ACF-OP and ACF-LP method,more calculation process can be found in the peper of Ackerberg et al.

### 3.2.2. Core Variables

This paper focused on the distortion of the labor ($\varphi_{lit}$) and capital ($\varphi_{kit}$) resources in Chinese furniture enterprises. Referring to the paper by Hsieh and Klenow (2009) [12], we calculated the distortion of the labor and capital resources of Chinese furniture enterprise and analyzed their impact on the TFP. We briefly explain the HK model studied by the Hsieh and Klenow as follows.

Suppose that enterprise *i* satisfies the Douglas production function:

$$Y_i = A_i L_i^\alpha K_i^\beta$$

$Y_i, A_i, L_i$, and $K_i$ represent the output, total factor productivity, labor factor input, and capital factor input, $Y_i$ is measured by the logarithm of the industrial added value of the enterprise, and $L_i$ is used as the logarithm of the labor average of the company throughout the year. $K_i$ is measured by the logarithm of the company's average annual balance of fixed assets. $\alpha$ and $\beta$ represent the elasticity of labor and capital, respectively, and we assume that $\alpha + \beta = 1$; the production function is assumed to be constant with respect to scale return.

For reference in the HK model, we used the actual prices of capital and labor for enterprises: $(1 + \varphi_{ki}) P_{ki}$, $(1 + \varphi_{li}) P_{li}$, respectively. Among these, $P_{ki}$ and $P_{li}$, respectively, represent the cost of capital and labor factors under complete competition in the factor market. We referred to the classic HK model used to measure the misallocation to set interest rate, and the interest rate did not affect the significance of the empirical result [12]. We assume that the price of product *i* is *P* and that the interest rate r of capital is 10%, which includes the actual interest rate of 5%, and the depreciation rate, also 5%. The price of labor is the average wage across the entire industry.

According to the company's profit maximization conditions:

$$\pi_i = PA_iL_i^{\alpha}K_i^{\beta} - (1 + \varphi_{ki})P_{ki}K_i - (1 + \varphi_{li})P_{li}L_i \tag{3}$$

The derivatives with respect to $L_i$ and $K_i$ are, respectively:

$$\frac{\partial \pi_i}{\partial K_i} = \partial PA_iL_i^{\alpha-1}K_i^{\beta} - (1 + \varphi_{ki})P_{ki} = 0$$

$$\frac{\partial \pi_i}{\partial L_i} = \beta PA_iL_i^{\alpha}K_i^{\beta-1} - (1 + \varphi_{li})P_{li} = 0$$

Then:

$$\varphi_{ki} = \frac{\alpha PY}{P_{ki}K_i} - 1$$

$$\varphi_{li} = \frac{\alpha PY}{P_{li}L_i} - 1$$

The export (EXP) behavior of an enterprise can be expressed in terms of the export delivery value, export proportion, or export pattern. [28–30]. If the export delivery value of the enterprise is used, it is impossible to compare and measure the actual export situation between enterprises of different scales. If the export pattern is used, the dummy variable may lead to a large amount of missing information, and cannot truly reflect the export levels of the enterprise. Therefore, we used the export proportion to measure the export behaviors of enterprises. The export proportion is the ratio of export delivery value of enterprises to the total value of industrial production. The proportion of exports can reflect both the export pattern and the true export level and ability of the enterprise. The calculation formula is as follows: EXP = export delivery value/total industrial production value.

### 3.2.3. Other Control Variables

Studies have shown that the micro-features of enterprises have a significant impact on TFP. In this paper, the following indicators were also selected as control variables.

We used the ratio of the added value of new products to the average number of laborers to measure enterprise innovation (R&D). R&D is an important factor in the improvement of enterprise productivity [31]. The more R&D innovation there is, the higher the efficiency of the enterprise. However, due to the lack of R&D data, we used the output value of new products instead. The calculation formula is as follows: R&D = *ln* (value of new product/number of employees+1).

To calculate the profitability of enterprise (PRO), we used the ratio of business operating profit to main business income. The profit level of an enterprise affects its investment. The higher the profit, the

higher its efficiency [6]. The calculation formula is as follows: PRO = *ln* (total profit/total operating income+1).

To calculate enterprise age (AGE), we used the company's statistical year minus the annual opening of the company. The longer a company has been established, the richer its resources, and the higher the TFP [5]. AGE is calculated as follows: enterprise age = the sample year – the year the company was listed.

For the state-owned or non-state-owned (NAT) variable, we used dummy variables to characterize ownership, with 1 for state-owned enterprises and 0 for non-state-owned enterprises. State-owned enterprises are generally less efficient and have lower TFPs [32].

The statistical descriptions of each variable in the model are shown in Table 1. The value of the TFP varies when it is calculated by different methods, and the distortion of labor and capital resources are affected by the TFP. The TFP results and distortion of the labor and capital resources are not shown in Table 1, as we will show them later.

**Table 1.** Statistical descriptions for the main variables.

| Variables | Variable Interpretation and Assignment | Mean | Standard Deviation | Expectation |
|---|---|---|---|---|
| TFP | total factor productivity (logarithm) | / | / | / |
| $\varphi_{kit}$ | the distortion of labor resources (logarithm) | / | / | + |
| $\varphi_{ki}$ | the distortion of capital resources (logarithm) | / | / | + |
| R&D | enterprise innovation (logarithm) | 0.2420 | 0.9721 | + |
| EXP | export behavior (export delivery value/total industrial production value) | 0.2968 | 0.4254 | - |
| NAT | state-owned or non-state-owned (yes = 1; no = 0) | 0.0422 | 0.2010 | - |
| PRO | the profitability of enterprise (logarithm) | −5.5629 | 4.8032 | + |
| AGE | enterprise age (years) | 7.8663 | 8.3198 | + |

*3.3. Descriptive Statistics*

3.3.1. Statistical Analysis of Total Factor Productivity of Furniture Enterprises

Based on a sample of more than 17,000 furniture enterprises, including data from 1999 to 2007, we used mainly the OP, ACF-OP, and ACF-LP methods to measure the TFP of Chinese furniture enterprises. We also measured the elasticity of labor and capital, and the results of different methods can be seen in Table 2. Referring to the practice of Yang Rudai (2015) [33], if the sum of the elasticity of labor and capital is less than 0.9, it will be defined as diminishing returns to scale. If the sum of the elasticity of labor and capital is between 0.9 and 1.0, it will be defined as the scale return. If the sum of the elasticity of labor and capital is greater than 1.0, it will be defined as increasing returns to scale. From Table 2, we know that the sum of the elasticity of labor and capital was less than 0.9 under the different methods. We can see that the Chinese furniture industry has diminishing returns to scale. The average TFP was approximately equal to 4, which is still relatively low compared to other industries.

**Table 2.** The elasticity of labor and capital of China's furniture industry.

| Method | $\alpha$ | Standard Deviation | $P > |z|$ | $\beta$ | S.D. | $P > |z|$ | $\alpha + \beta$ | TFP |
|---|---|---|---|---|---|---|---|---|
| OP | 0.5267 | 0.0137 | 0.000 | 0.2607 | 0.0236 | 0.000 | 0.7874 | 3.8885 |
| ACF-OP | 0.6108 | 0.0177 | 0.000 | 0.1817 | 0.0210 | 0.000 | 0.7924 | 4.0077 |
| ACF-LP | 0.7601 | 0.0621 | 0.000 | 0.0494 | 0.0300 | 0.100 | 0.8095 | 4.4628 |

Table 3 shows the changes in the TFP of the furniture industry based on the ACF-OP method. As the results of the three methods were similar, we only show the results of the calculation using the ACF-OP method. The number of furniture enterprises in different sub-sectors varied, with wooden furniture enterprises accounting for the largest portion, followed by metal furniture companies, and bamboo, rattan furniture, and plastic furniture enterprises. There were no obvious differences in the TFP of different sub-sectors, but the TFP of all sub-sectors showed an upward trend, and the wood furniture industry had a relatively large increase. The total factor productivity of the wood furniture industry increased from 3.4227 in 1999 to 4.3374 in 2007.

**Table 3.** Annual TFP of different sub-sectors based on the ACF-OP method.

| Industry | Name | 1999 | 2000 | 2001 | 2002 | 2003 | 2005 | 2006 | 2007 |
|---|---|---|---|---|---|---|---|---|---|
| 2110 | TFP | 3.4227 | 3.6628 | 3.6310 | 3.7116 | 3.8024 | 4.0225 | 4.1700 | 4.3374 |
|  | N | 895 | 932 | 1025 | 1106 | 1271 | 1858 | 2177 | 2463 |
| 2120 | TFP | 3.6432 | 4.1247 | 4.0357 | 4.1530 | 3.8556 | 3.9511 | 3.9054 | 4.0378 |
|  | N | 21 | 20 | 17 | 17 | 25 | 46 | 62 | 63 |
| 2130 | TFP | 3.6569 | 3.7707 | 3.7885 | 3.8748 | 3.9850 | 4.1473 | 4.3023 | 4.3941 |
|  | N | 197 | 229 | 296 | 329 | 426 | 672 | 792 | 864 |
| 2140 | TFP | 3.7359 | 4.4027 | 4.0184 | 3.6526 | 3.9944 | 4.4360 | 4.3417 | 4.3293 |
|  | N | 6 | 10 | 12 | 12 | 16 | 49 | 51 | 65 |
| 2190 | TFP | 3.6355 | 3.8660 | 3.7590 | 3.7823 | 3.8301 | 4.1394 | 4.2686 | 4.4369 |
|  | N | 96 | 132 | 171 | 177 | 200 | 307 | 389 | 505 |

### 3.3.2. Statistical Analysis of the Distortion of Labor and Capital Resources in Furniture Enterprises

Table 4 shows the TFP of Chinese furniture enterprises based on the HK model (Hsieh et al. 2009), calculated using the OP, ACF-OP, and ACF-LP methods. Under the three methods, the distortion in labor resources was larger than that of capital resources, and both exhibited an upward trend. This could be because China's economy was in an overheated state during 2004–2007. To control the rate of economic growth, the government has implemented a series of policies to control the factor market and strengthened environmental regulations.

**Table 4.** The distortion of labor and capital resource in furniture enterprises.

| Method | OP | | ACF-OP | | ACF-LP | |
|---|---|---|---|---|---|---|
| Year | $\varphi_{kit}$ | $\varphi_{lit}$ | $\varphi_{kit}$ | $\varphi_{lit}$ | $\varphi_{kit}$ | $\varphi_{lit}$ |
| 1999 | 2.2425 | 5.6470 | 1.8923 | 5.6407 | 0.5815 | 6.0134 |
| 2000 | 2.1865 | 6.8023 | 1.8373 | 6.8023 | 0.5256 | 7.1687 |
| 2001 | 2.2176 | 5.5427 | 1.8684 | 5.5427 | 0.5567 | 5.9090 |
| 2002 | 2.6292 | 5.6504 | 2.2799 | 5.6504 | 0.9683 | 6.0167 |
| 2003 | 3.1125 | 5.6738 | 2.7633 | 5.6738 | 1.4516 | 6.0401 |
| 2005 | 2.4745 | 6.0337 | 2.1252 | 6.0337 | 0.8136 | 6.4000 |
| 2006 | 2.6346 | 6.0597 | 2.2854 | 6.0597 | 0.9737 | 6.4261 |
| 2007 | 2.6164 | 6.2818 | 2.2671 | 6.2818 | 0.9555 | 6.6482 |

## 4. Empirical Results and Analysis

### 4.1. The Regression Results of the Benchmark Model

Table 5 shows the final regression results based on the two-way fixed-effects model obtained using the OP, ACF-OP, and ACF-LP method to calculate the TFP as the dependent variable for regression. The regression results of the model were better, with R-squared values of approximately 0.8. This indicates that the selected variables can be interpreted as the dependent variables, and that these variables are significant.

**Table 5.** The regression results with two-way fixed-effects model.

| Variables | *ln* OP | *ln* ACF-OP | *ln* ACF-LP |
|:---:|:---:|:---:|:---:|
| $\varphi_{kit}$ | 0.0846 *** | 0.0640 *** | 0.0208 *** |
| | (0.0017) | (0.0014) | (0.0011) |
| $\varphi_{lit}$ | 0.1962 *** | 0.2157 *** | 0.2126 *** |
| | (0.0032) | (0.0033) | (0.0025) |
| EXP | −0.0030 *** | −0.0032*** | −0.0025 ** |
| | (0.0011) | (0.0010) | (0.0013) |
| R&D | 0.0041 *** | 0.0031 *** | 0.0034 *** |
| | (0.0011) | (0.0011) | (0.0009) |
| PRO | 0.0002 *** | 0.0001 *** | 0.0001 *** |
| | (0.0004) | (0.0004) | (0.0003) |
| AGE | 0.0003 *** | 0.0004 *** | 0.0001 *** |
| | (0.0003) | (0.0003) | (0.0002) |
| NAT | −0.2339 *** | −0.2064 *** | −0.1749 *** |
| | (0.0244) | (0.0219) | (0.0171) |
| Constant | 1.0904 *** | 1.1150 *** | 1.3068 *** |
| | (0.0088) | (0.0076) | (0.0067) |
| Time effects | control | control | control |
| Individual effects | control | control | control |
| *R*-squared | 0.7916 | 0.8192 | 0.8436 |
| Observations | 17610 | 17610 | 17610 |

Note: t statistics in parentheses, * $p < 0.1$, ** $p < 0.05$, *** $p < 0.01$.

The coefficient of EXP was negative, and the significance test at the 1% level indicates that the relationship between the exports and TFP of the enterprise was negative. We can identify an export–productivity paradox in the Chinese furniture industry, which supports hypothesis H1. There is no "export learning effect" or "export self-selection effect" in China's furniture industry. As the Chinese furniture industry is mainly based on processing trade and labor-intensive enterprises, products are used for export, and the cost for export is low, so it is easy to enter foreign markets. Moreover, Chinese furniture enterprises use China's abundant labor resources, which results in the competitive pressure in the domestic market being larger than that of the competitive pressure to enter foreign markets. Therefore, the Chinese furniture industry exhibits the export–productivity paradox.

The coefficients of $\varphi_{kit}$ t and $\varphi_{lit}$ were positive, and the significance test at 1% indicates that the distortions of labor and capital resources were positively correlated with the TFP of enterprises, which supports hypothesis H2. We can determine whether furniture enterprises face misallocation in the factor market because such misallocation causes adverse effects. Furniture enterprises must improve their competitiveness by improving the TFP to guarantee their survival. The "reverse mechanism" can be used to improve the TFP. At the same time, we found that the impact of distortion in labor resources on the TFP was larger than that of distortion of capital resources.

The R&D coefficient was positive, and the 1% level of significance indicates that there was a positive correlation between R&D innovation and TFP. The TFP of enterprises with R&D innovation were higher than in the case of those without R&D innovation. R&D innovation is conducive to improvements in the TFP of furniture enterprises. Hence, we can improve the TFP of furniture enterprises by increasing R&D investment and improving technical capabilities. The coefficient of PRO was positive, and the significance test at the 1% level indicates that there was a significant positive correlation between profit levels and TFP of enterprises. As profits continue to increase, an enterprise will be able to increase its investment and enhance its TFP. The coefficient of AGE was positive, and the significance test at the 1% level indicates that there was a significant positive correlation between a company's age and its TFP. As the company's lifetime in the market increases, its management and production will continue to improve, so the company is more likely to increase its TFP. The coefficient of NAT was negative, and the 1% level of the significance indicates that the TFP of state-owned furniture enterprises was low, but that of non-state-owned furniture enterprises was higher. Non-state-owned

enterprises are more efficient, actively participate in market competition, and constantly improve their TFP.

*4.2. Moderating Effects*

This section mainly discusses the relationship between exports, misallocation, and TFP. Furniture enterprises are mainly labor-intensive enterprises, and the degree of the distortion of labor resources is higher than that of capital resources. Therefore, we mainly discuss the distortion of labor resources in this section. We analyzed the intersection between the export situation and the distortion of labor resources and added it to the regression model to explore whether labor distortion had a moderating effect on exports.

Table 6 shows the regression results for this moderating effect. The 1% level of significance indicates that the distortion of labor resources had a positive impact on exports. Adding the interaction item had a small effect on the ACF-LP method, but the value was still positive. The coefficient of the distortion of labor resources became positive, which further supports it having a positive moderating effect on exports. This result coincides with the development of China's furniture industry. Furniture enterprises are mostly processing trade-oriented and labor-intensive enterprises, which rely on intensive investment in labor resources. As China's "demographic dividend" gradually disappears, labor costs continue to rise, and furniture enterprises must reduce their dependence on labor factors. Changes in the enterprise environments will continue to stimulate businesses to improve production efficiency. Hence, furniture enterprises will promote improvements to the TFP further through a forced mechanism to ensure their survival. This supports hypothesis H3.

**Table 6.** The regression results of moderating effect.

| Variables | ln OP | ln ACF-OP | ln ACF-LP |
|---|---|---|---|
| $\varphi_{kit}$ | 0.0846 *** | 0.0640 *** | 0.0206 *** |
| | (0.0017) | (0.0014) | (0.0011) |
| $\varphi_{lit}$ | 0.1960 *** | 0.2155 *** | 0.2139 *** |
| | (0.0035) | (0.0035) | (0.0024) |
| EXP | 0.0038 *** | 0.0030 *** | −0.0025 *** |
| | (0.0007) | (0.0006) | (0.0012) |
| $\varphi_{lit} \times$ EXP | 0.0027 *** | 0.0026 *** | 0.0001 *** |
| | (0.0010) | (0.0009) | (0.0000) |
| R&D | 0.0030 *** | 0.0021 * | 0.0042 *** |
| | (0.0012) | (0.0011) | (0.0008) |
| PRO | 0.0007 *** | 0.0006 *** | 0.0002 *** |
| | (0.0004) | (0.0003) | (0.0003) |
| AGE | 0.0001 *** | 0.0003 *** | 0.0001 *** |
| | (0.0003) | (0.0003) | (0.0002) |
| NAT | −0.2273 *** | −0.2001 *** | −0.1741 *** |
| | (0.0243) | (0.0218) | (0.0170) |
| Constant | 1.0854 *** | 1.1102 *** | 1.3059 *** |
| | (0.0089) | (0.0076) | (0.0066) |
| Time effects | control | control | control |
| Individual effects | control | control | control |
| *R*-squared | 0.7957 | 0.8233 | 0.8447 |
| Observations | 17610 | 17610 | 17610 |

Note: t statistics in parentheses, * $p < 0.1$, ** $p < 0.05$, *** $p < 0.01$.

**5. Results and Discussion**

Based on data from the China Industrial Enterprise Database from 1999 to 2007, this paper analyzed the impact of both exports and misallocation on the TFP of Chinese furniture enterprises. In the background of China's labor resources changes and the dramatic changes in the international trade situation, Chinese furniture enterprises relying on cheap labor as their main competitive advantage

need to transform to rely on the TFP to transform and upgrade. However, we found that existing research does not accurately answer the impact of exports and misallocation on the TFP of furniture enterprises through literature review. Firstly, based on the Olley and Pakes (OP), Ackerberg, Caves, and Frazer–OP (ACF-OP), and ACF–Levinsohn and Petrin (ACF-LP) methods, we calculated the TFP of Chinese furniture enterprises. We believe that Chinese furniture enterprises were in the stage of diminishing returns to the overall scale during 1999–2007. There were no obvious differences in the TFP of different sub-sectors, but the TFPs of all sub-sectors showed an upward trend, and the wood furniture industry had a relatively large increase. Then, based on the HK model, we calculated the misallocation level of Chinese furniture enterprises, the distortion in labor resources was larger than that of capital resources, and both exhibited an upward trend. In the end, a two-way fixed-effects model was used to analyze the interactions between exports, misallocation, and the TFP.

The empirical results were as follows, First, there is an export–productivity paradox in the Chinese furniture industry. The TFP of export enterprises is lower than that of non-export enterprises, and there is no "self-selection effect" or "export learning effect". The Chinese furniture industry is mainly based on processing trade and labor-intensive enterprises. Processing trade is characterized by "two heads outside" and products are used for export, thus lowering the productivity of the entire export enterprise. The TFP of Chinese furniture export enterprises is lower than that of non-export enterprises. However,furniture enterprises are located at the low end of the industrial value chain, the situation of the export–productivity paradox is only a short-term situation, the export learning effect should be long-term. Second, contrary to our general understanding, we found that the misallocation forces Chinese furniture enterprises to increase their TFP. When furniture enterprises face misallocation, enterprises must increase the TFP to make up for the increase in production costs brought by the misallocation, otherwise the enterprises will be eliminated by the market. However, this situation is a short-term phenomenon; we can look at Figure 1, the horizontal axis is the square of the distortion of labor resource, the vertical axis is the TFP. We also discussed interaction between the misallocation and TFP, and the relationship between them was an inverted U-shape. In the short term, the interaction between them is positive, the misallocation may enhance the TFP of the furniture enterprises, the interaction between them is negative in the long term. However, the distortion of labor resources can regulate the negative impact of exports on the TFP. The distortion of labor resources can reduce the negative impact of exports on the TFP of enterprises and improve the TFP of furniture enterprises through the forced mechanism.

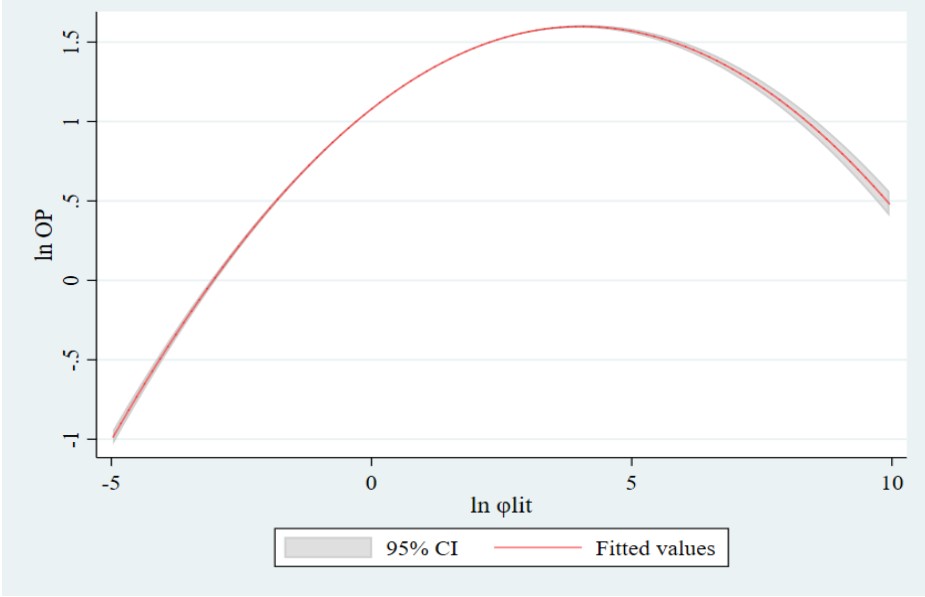

**Figure 1.** The interaction between the TFP and the $\varphi_{lit}$ in different periods.

Our research seems to be able to draw the conclusion that Chinese furniture enterprises can automatically achieve industrial upgrading based on changes in market conditions, the government should take back the "visible hand" and promote the economy to achieve market-oriented development. Our empirical results also support this conclusion, we can see that furniture enterprises can improve the TFP according to their own characteristics and the changes in the supply of factors to the market. The TFP of state-owned enterprises is also significantly lower than that of private enterprises. At present, China is also further deepening market-oriented reforms, emphasizing the development of a high-quality and efficient market economy, innovating the reform ideas of state-owned enterprises, and attaching importance to the role of market players in order to form an open market economic system. However, we cannot ignore the important role of R&D in the improvement of the TFP. We found that furniture enterprises with R&D investment have significantly higher TFP than those without R&D investment. Processing trade enterprises generally have low R&D and innovation capabilities, the government should increase support for furniture enterprises to innovate, thus helping furniture enterprises to find new driving forces as the demographic dividend weakens.

**Author Contributions:** C.X. and J.G performed calculation, analyzed the data and wrote the paper. B.C. verified and solidified the argument, edited the paper and drafted the conclusions. Y.L. revised the manuscript during the whole writing process. All authors contributed to the drafting of the article and read the final manuscript.

**Funding:** This research is supported by grants from by the National Natural Science Foundation of China (Grant No. 71873016), the National Natural Science Foundation of China (Grant No.71974186, 71690242), the National Key Research and Development Program of China (Grant No.2016YFA0602500) and the Strategic Priority Research Program of Chinese Academy of Sciences (Grant No. XDA20100104).

**Conflicts of Interest:** The authors declare no conflict of interest.

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
