# Peer review of "Exports, Misallocation, and Total Factor Productivity of Furniture Enterprises"

_sustainability, doi:10.3390/su11184892_

Round 1

Reviewer 1 Report

WARNING:
Lines 168 on: all references are missing, I cannot comment on that. The issue carries on the entire paper. Hence my comments are partial. Please correct.

SUMMARY

The paper investigates empirically the relation between TFP and exports of Chinese furniture firms.
Three main findings:
1) Confirm the existence of an “export-productivity paradox”
2) Misallocation spurred TFP improvements, rather than hamper them
3) Misallocation by moderating exports may improve TFP

MAIN COMMENTS

Abstract:
Should be made clearer.
The "forced mechanism" should be explained better.

Lines 85-86: The finding that furniture exporters are less productive than non-exporters needs to be qualified. Is it a ceteris paribus comparison? This is important and comes naturally from the discussion done few lines above on value chains and processing trade. To what extent this “export-productivity paradox” depends on the fact that exporters focus on some low value-added activities, while non-exporters also perform other types of activities? The question is whether exporters are more or less productive of non-exporting firms with similar characteristics and performing similar activities.

Line 87-88: The authors rightly suggest that the lower productivity of exporters is probably mostly due to a composition effects stemming from the fact that exporters are more involved in processing trade, assembly and other low value added activities. This partially answers to my previous comment. The authors should try to be clearer on this point.

Line 88: Again the definition of misallocation is vague.

Line 96: The "forced mechanism" should be explained better as well as the authors' contribution.

Line 109: The sentence is unclear and it seems wrong because TFP refers to a measure of productivity, which is a static concept, while growth is dynamic concept (e.g. growth in productity).

The discussion on value chains and processing trade as explanations for the lower productivity of exporters, while reasonable, also raises some questions. It is renowned that China benefited from export led growth in the last two decades and that this was mostly due precisely to processing trade, special economic zones and similar policies. It is usually argued that processing trade and value chain participation brought about some positive spillovers and knowledge transfer to the involved firms and to the rest of the economy, thus improving TFP and contributing to country development. From this point of view, this is at odds with the evidence that export (processing trade) hampers TFP because if this was the main effect, then China would have done better to close its economy. Similarly, if misallocation and other distortions have a positive effect on productivity, China (and other countries) should be willing to heavily distort their economies to improve productivity. This is clearly paradoxical, so these positive effects must be of second order. I speculate that the two views may be reconciled by the different perspectives as the latter (negative) effect of exports is probably due to a short run composition effect, while the spillover effect is mostly a long run effect. Anyway, I advice the authors to discuss the issue more in depth.

Is the "export-productivity paradox" found in other countries involved in international trade through low value added phases and/or processing trade/assembly?

Lines 208 on: not clear what you mean by "samples".

Line 213-217: it is not clear why 2004 must be excluded.

Line 213-217: Is the imputation of missing value added data with the provided formula robust or are the authors introducing a distortion? To be sure, the authors may want to impute value added even for firms for which it is reported and compare the imputed and the reported numbers.

Lines 203-244: Please revise the paragraph as the description of the different methods seems quite imprecise. Also, I think DEA stands for Data Envelopment Analysis.

Lines 269: the assumptions on the interest rate must be motivated.

Line 275: there seems to be some issues with the formulas. Please check.

Line 278: the authors should explain better what are the "phis". (perhaps this is an issue due to the missing references, which makes the text hard to read)

Line 298: is the PRO index calculated from the balance sheets?

SUGGESTIONS

In the first part of the paper, it would be interesting to provide some figures comparing TFP and/or its growth (or cost of labor or other productivity measures) across major furniture exporting countries.

Lines 49-51: it might be interesting to expand the discussion on the value chains.

MINOR COMMENTS

The hypotheses H1, H2 etc. could be emphasized through an appropriate text formatting to facilitate the reader.

Line 44: clarify how big is two pages.

Line 47: should be updated.

Line 54: the authors should more precisely refer to the so called "new new trade theory". I am not sure [3] can be classified as such.

Define the acronyms the first time you introduce them (e.g. TFP, OEM etc. are undefined).

I strongly recommend a check of the English.

Author Response

Detailed Response Letter

Dear my reviewer,

We are very grateful for your comments on our manuscript entitled “Exports, Misallocation and Total Factor Productivity of Furniture Enterprises” (Sustainability-562210). The feedback has been valuable to improve the manuscript. We have thoroughly revised the manuscript according to every comment and suggestion. The changes made to the manuscript are detailed below.The specific content is in the attachment.

We hope that we have thus adequately addressed your comments and our revised manuscript now meets the standards of the Sustainability.

Once again, we are very grateful to your comments and suggestions.

Sincerely,

Chang Xu on behalf of all co-authors

Reviewer 2 Report

The paper traits a very interesting topic highlighting the impact and interaction of both exports and misallocation on the TFP. However, the paper suffers because of the lack of scientific consistency.
The introduction need to present into a more concise manner the relevance of the topic and the motivation of the study. The section of literature review presenting the most relevant studies in the field is totally missing and the sections of empirical results and conclusions need to be more detailed.

Within the paper there is a huge issue with the equation editor that need to be solved.

Author Response

(The authors gave the same response as above.)

Round 2

Reviewer 1 Report

The paper has improved and the authors answered appropriately to most comments.

I still have some minor concerns.

Model (2) does not include EXP as a single regressor EXP, which is instead included later in the regression tables. Please correct.

It is unclear whether TFP is estimated for the entire sector or separately for each sub-sector. In the former case, Table 2 should also report the standard errors and the p-values.

What are the reasons to not try an interaction effect also between capital distortions and EXP? What would be its interpretation? This should be either motivated or showed.

Regression tables should include the number of observations.

Figure 1 can be made clearer.

Author Response

Detailed Response Letter

Dear my reviewer,

We are very grateful for your latest comments on our manuscript entitled “Exports, Misallocation and Total Factor Productivity of Furniture Enterprises” (Sustainability-562210). The feedback has been valuable to improve the manuscript. We have thoroughly revised the manuscript according to every comment and suggestion. The changes made to the manuscript are detailed below.

We hope that we have thus adequately addressed your comments and our revised manuscript now meets the standards of the Sustainability.

Once again, we are very grateful to your comments and suggestions.

Sincerely,

Chang Xu on behalf of all co-authors

Reviewer 2 Report

The paper has been significantly improved.Therefore, I recommend the publication.

Author Response

Detailed Response Letter

Dear my reviewer,

We are very grateful for your comments on our manuscript entitled “Exports, Misallocation and Total Factor Productivity of Furniture Enterprises” (Sustainability-562210). Thank you for your guidance and help. Without your help, we will not upgrade our papers. We learn a lot from your comments and suggestions. Thank you very much again

Once again, we are very grateful to your comments and suggestions.

Sincerely,

Chang Xu on behalf of all co-authors
